# Differentiation between myopericarditis and acute myocardial infarction on presentation in the emergency department using the admission C-reactive protein to troponin ratio

**Simcha R. Meisel** [ID]\*, **Hamuda Nashed, Randa Natour, Rami Abu Fanne, Majdi Saada, Naama Amsalem, Carmel Levin, Ofer Kobo, Aaron Frimerman, Yaniv Levi, Jameel Mohsen, Avraham Shotan, Ariel Roguin, Michael Kleiner-Shochat**

Heart Institute, Hillel Yaffe Medical Center, Hadera, Israel

\* meisel@hymc.gov.il

## Abstract

### Background

The treatment of myopericarditis is different than that of acute myocardial infarction (AMI). However, since their clinical presentation is frequently similar it may be difficult to distinguish between these entities despite a disparate underlying pathogenesis. Myopericarditis is primarily an inflammatory disease associated with high C-reactive protein (CRP) and relatively low elevated troponin concentrations, while AMI is characterized by the opposite. We hypothesized that evaluation of the CRP/troponin ratio on presentation to the emergency department could improve the differentiation between these two related clinical entities whose therapy is different. Such differentiation should facilitate triage to appropriate and expeditious therapy.

### Methods

We evaluated the CRP/troponin ratio on presentation among patients consecutively included in a large single center registry that included 1898 consecutive patients comprising 1025 ST-elevation myocardial infarction (STEMI) patients, 518 Non-STEMI (NSTEMI) patients, and 355 patients diagnosed on discharge as myopericarditis. CRP and troponin were sampled on admission in all patients and their ratio was assessed against discharge diagnosis. ROC analysis of the CRP/troponin ratios evaluated the diagnostic accuracy of myopericarditis against all AMI, STEMI, and NSTEMI patients.

### Results

Median admission CRP/troponin ratios were 84, 65, and 436 mg×ml/liter×ng in STEMI, NSTEMI and myopericarditis groups, respectively (p<0.001) demonstrating good differentiating capability. The Receiver-operator-curve of admission CRP/troponin ratio for diagnosis of myopericarditis against all AMI, STEMI, and NSTEMI patients yielded an area-under-the

**Data Availability Statement:** All relevant data are within the paper and its Supporting information files.

**Funding:** The authors received no specific funding for this work.

**Competing interests:** The authors have declared that no competing interests exist.

curve of 0.74, 0.73, and 0.765, respectively. CRP/troponin ratio>500 resulted in specificity exceeding 85%, and for a ratio>1000, specificity>92%.

## Conclusion

The CRP/troponin ratio can serve as an effective tool to differentiate between myopericarditis and AMI. In the appropriate clinical context, the CRP/troponin ratio may preclude further evaluation.

## Introduction

It is frequently difficult to differentiate on clinical grounds between myopericarditis and acute myocardial infarction (AMI) as the cause of an acute cardiac event [1] since both conditions present with acute chest pain, electrocardiographic changes and elevated troponin level [2, 3]. Occasionally, myopericarditis may mimic AMI and present acutely with typical ischemic symptoms, while the electrocardiogram may demonstrate ischemic changes or even simulate an ST-segment elevation myocardial infarction (STEMI). In fact, even echocardiography performed in patients with myopericarditis may demonstrate segmental wall motion abnormalities typical of an AMI and not show the expected pattern of diffuse hypokinesis [4] observed as patchy involvement on CMR [5]. In light of the specific diagnostic tests required and disparate therapeutic approach in these conditions, it is imperative to establish the diagnosis early, hopefully in the emergency department, in order to provide expeditiously the appropriate therapy. This may require further imaging or invasive tests. Any biochemical test that, in the appropriate clinical setting, could assist in distinguishing between myopericarditis and AMI would be helpful.

Troponin and C-reactive protein (CRP), the universal inflammatory factor [6], are generally elevated in both AMI and in myopericarditis. However, the primary pathological process in patients with myopericarditis is inflammation, whatever the etiology [1]. Therefore, it is typically associated with high levels of CRP [7] but with a proportionally smaller increase in the level of troponin released from damaged cardiomyocytes [8]. On the other hand, STEMI and to a lesser degree NSTEMI, with their attendant myonecrosis are associated with higher levels of troponin, while CRP level is also elevated in these conditions but not to the same extent as troponin (Fig 1). Hence, we aimed to test the hypothesis that the ratio of CRP to troponin level, expected to be high in the former and low in the latter, would improve our ability to differentiate on patient presentation between these often-similar clinical entities.

## Methods and patients

The current study was based on our local heart institute registry that included all consecutive patients admitted between January 2011 and April 2017 with any type of chest pain with electrocardiographic changes and discharged after evaluation and therapy with either the diagnosis of AMI or myopericarditis. Patients with the discharge diagnosis of pulmonary embolism, decompensated heart failure, aortic dissection, and sepsis were excluded.

According to hospital routine, CRP and troponin are sampled on presentation to the emergency department in all patients. Admission CRP/troponin ratio was calculated for all patients and used for analysis of its ability to differentiate between patients diagnosed with myopericarditis and all AMI patients or between STEMI or NSTEMI patients only. The local institutional

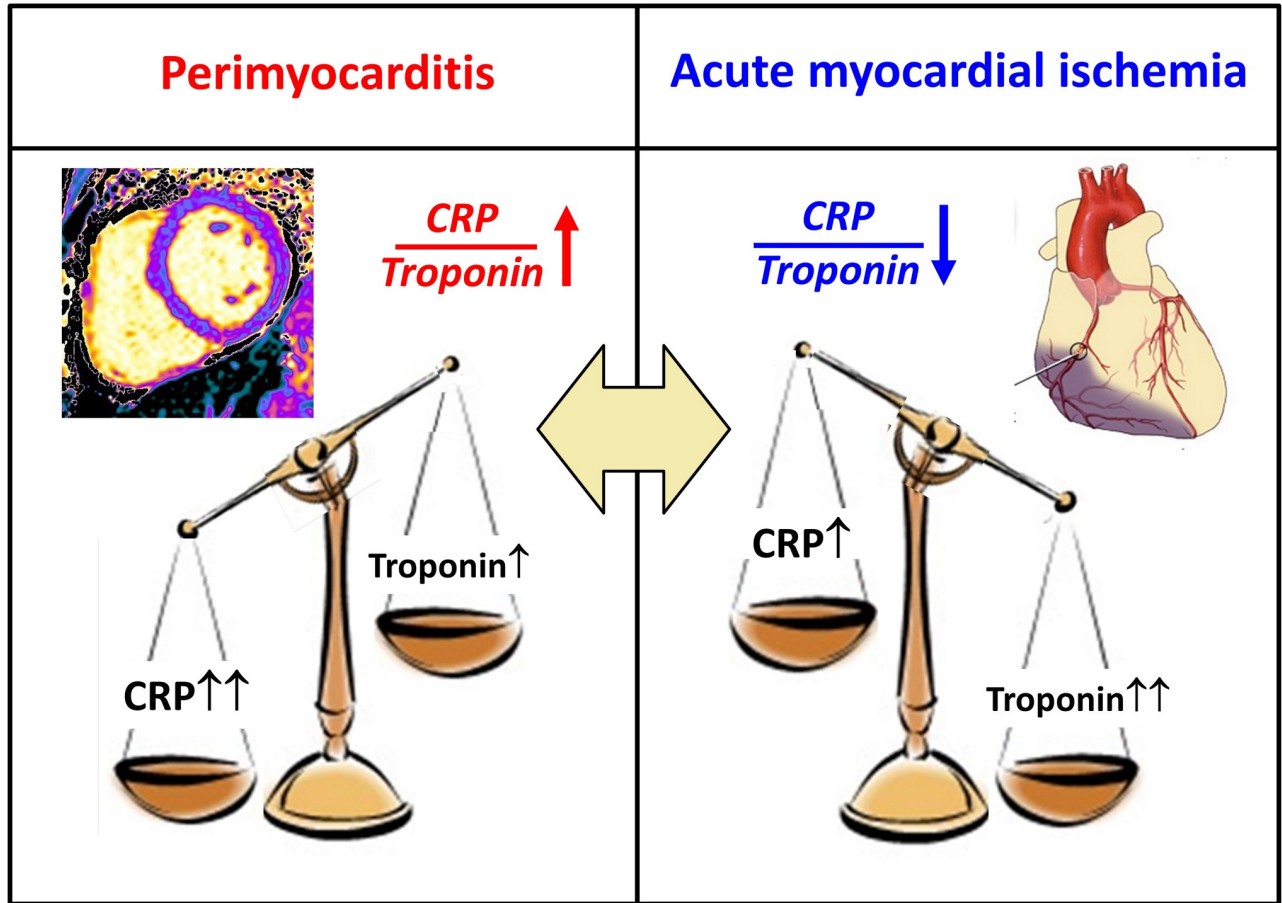

**Fig 1. Central figure showing study rationale.** The primary pathologic process in myopericarditis is inflammation associated with high CRP and relatively low-range but higher than normal troponin concentrations, while myocardial infarction is characterized by elevated troponin with a relatively lower increase in CRP levels. The CRP/troponin biomarker will amplify these differences.

review board approved this retrospective registry-based analysis of prospectively collected data. Informed consent was waived due to the retrospective and diagnostic nature of the study. The study required no external funding.

*Diagnosis* myopericarditis is an inflammatory disease involving both the pericardium and the myocardium [1, 4]. Apart from its typical or atypical clinical presentation, medical circumstances such as a recent febrile illness or other conditions and elevated CRP and troponin, myopericarditis is mostly diagnosed by performing a CMR, or by excluding hemodynamically-significant obstructive coronary disease. In general, myopericarditis was diagnosed in the current study by the following: (1) Typical or atypical chest pain; (2) Elevated CRP; (3) Electrocardiographic changes, and (4) Either a CMR demonstrating the pathological pattern compatible with myopericarditis, or coronary or cardiac CT angiography excluding obstructive coronary disease. Usually, the echocardiogram showed segmental wall motion abnormalities and at times a diffuse hypokinetic pattern, with or without some degree of pericardial fluid. Frequently there was a history of recent febrile disease but this was not obligatory. AMI was diagnosed according to the ESC guidelines for NSTEMI [9] and STEMI [10] based on ischemic symptoms, typical dynamic changes on the electrocardiogram, and elevated troponin levels, in light of coronary artery anatomy on angiography.

CRP was measured in the current study by the particle enhanced immuno-turbidimetric assay with the upper normal reference level of 5 mg/L as described (CRPL3, C-Reactive Protein Gen.3, ref No.05172373 190, Roche Diagnostics GmbH, Mannheim, Germany). Troponin T was measured in this study by the Sandwich principle with normal troponin T concentration of < 0.014 ng/ml (Troponin T hs, STAT ref No. 05092728. System information for COBASe 411 analyzer. Roche Diagnostics GmbH, Mannheim, Germany). In the present study, the CRP was represented as mg/liter, while troponin levels were expressed as ng/ml in order to yield integer values for CRP/troponin. If troponin concentrations are expressed as ng/liter and used in the CRP/troponin ratio formula the result should be multiplied by 1000 to yield values as used in this paper. The submission contains the minimal data set including the values used to calculate the diagnostic accuracy of the CRP/troponin ratio to diagnose myopericarditis against AMI, STEMI and NSTEMI, as well as the prevalence of perimyocarditis within the CRP/troponin ratio deciles as percentage of total study population.

*Statistics* Continuous variables were expressed as mean ± SD or median (interquartile range) as appropriate. Categorical data were presented as absolute numbers and percent frequencies, with differences evaluated by the chi-square test or Fisher exact test as suitable. The student *t* test and Mann-Whitney *U* tests were used to analyze differences between means.

Multivariate logistic regression model was applied to predict myopericarditis disease with adjustment to the independent parameters of CRP/troponin ratio, age and creatinine. We assessed CRP/troponin ratio and patient age in the model by their quartiles: 29, 29–117, 117–455, >455 and <50, 50–59, 59–60, >69 years, respectively.

To assess the ability of the CRP/troponin ratio to predict myopericarditis against AMI, STEMI, or NSTEMI, their areas under the receiver-operating curves (ROC) with corresponding confidence intervals (CI) were calculated. Sensitivity, specificity, positive (PPV) and negative predictive (NPV) values were calculated for several optional CRP/troponin values including the optimal value of the Youden J statistic. In addition, we have calculated the positive and negative likelihood ratios. The former reflects the likelihood of obtaining a positive test result among patients with myopericarditis compared to a positive result in those without myopericarditis. The latter ratio reflects the likelihood of a negative result among myopericarditis patients compared to the likelihood of a negative result among those without the condition. 2-sided p value of <0.05 was considered statistically significant. All statistical analyses were performed using SPSS version 21.0 (Chicago, Illinois).

## Results

The present study included 1898 patients comprising 1025 STEMI patients, of whom 232 patients did not undergo primary reperfusion due to late arrival, refusal or associated critical medical conditions, 518 NSTEMI patients and 355 patients diagnosed with myopericarditis (Table 1). All STEMI and NSTEMI patients were hospitalized in the cardiac care unit from January 2011 to April 2017, and 1509 of 1543 AMI patients underwent coronary angiography. All patients diagnosed as AMI in the medical wards were transferred during their hospital stay to the cardiac care unit for coronary angiography and therapy and were thus accounted for under the cardiac unit population. The registry also included 169 patients admitted to the cardiac care unit with the diagnosis or suspicion of AMI and discharged with the diagnosis of myopericarditis, and 186 patients admitted to the medical wards during the same period and discharged with the diagnosis of myopericarditis. Of the latter, 60 patients underwent coronary angiography that ruled out clinically significant coronary disease in 56 patients and lead to an intervention in 4 patients. Myopericarditis patients were younger (50.3±19.5 years) than the STEMI patients (60±13, p<0.001) and the NSTEMI patients (63±12, p<0.001) patients. The

**Table 1. The demographic and basic clinical characteristics including CRP and troponin levels and their ratio among study patient groups according to the diagnosis at discharge.**

| Variable | STEMI (n = 1025) | Non STEMI (n = 518) | Myopericarditis (n = 355) | p-value |
|---|---|---|---|---|
| Age | 60±12.7 | 63±11.9 | 50.3±19.5 | p<0.001[1,2,3] |
| Gender (M) | 835 (81.5%) | 401 (77.4%) | 237 (63.7%) | p<0.0001[1,2,3] |
| S/P Congestive heart failure | 57 (5.6%) | 44 (8.5%) | 40 (11%) | p<0.05[1,2] |
| S/P Myocardial infarction | 240 (23.5%) | 172 (33.3%) | 29 (8%) | p<0.001[1,2,3] |
| S/P percutaneous coronary intervention | 276 (27.0%) | 168 (32.6%) | 41 (11%) | p<0.001[2,3] |
| S/P cerebrovascular accident | 59 (5.8%) | 61 (11.8%) | 22 (6.0%) | p<0.01[1,3] |
| Hypertension | 605 (59.1%) | 360 (69.5%) | 162 (44.5%) | p<0.001[1,2,3] |
| Hyperlipidemia | 636 (62.1%) | 377 (72.8%) | 133 (36.5%) | p<0.001[1,2,3] |
| Diabetes mellitus | 253 (25.3%) | 160 (31.1%) | 73 (30.7%) | p = 0.017[1] |
| Peripheral arterial disease | 46 (4.5%) | 49 (9.5%) | 30 (8.2%) | P<0.05[1,2] |
| Chronic renal failure | 97 (9.5%) | 75 (14.5%) | 55 (15.2%) | P<0.05[1,2] |
| Smoking | 394 (38.7%) | 232 (44.9%) | 239 (65.7%) | p<0.001[1,2,3] |
| ST-elevations | 985 (96.1%) | 10 (1.9%) | 111 (30.6%) | p<0.001[1,2,3] |
| ST-depressions | 9 (0.9%) | 168 (32.4%) | 17 (4.7%) | p<0.001[1,2,3] |
| T-wave inversions | 9 (0.9%) | 118 (22.8%) | 60 (16.5%) | p<0.001[1,2,3] |
| Death during admission | 50 (4.9%) | 13 (2.5%) | 13 (3.6%) | p = 0.029[1] |
| Segmental wall motion Score Index | 1.65±0.46 | 1.44±0.44 | 1.34±0.48 | p<0.01[1,2,3] |
| CRP median [25–75] | 7.2 [3.1–21.7] | 6.3 [2.8–13.3] | 32 [10.3–86.8] | p<0.001[1,2,3] |
| Early cTn median [25–75] | 0.07 [0.02–0.48] | 0.07 [0.02–0.22] | 0.04 [0.01–0.3] | p<0.05[23] |
| CRP/cTn ratio median [25–75] | 84.1 [20.9–308] | 65 [25.6–213] | 436 [93–3223] | p<0.001[2,3] |

[1] p- significance of difference between NSTEMI vs. STEMI;

[2] p- between NSTEMI vs. pericarditis; and

[3] p- between STEMI vs. pericarditis. CRP- C-reactive protein.

percentage of male patients among the STEMI patients was similar to that in NSTEMI patients but higher than that in the myopericarditis group (81.5% and 77% vs. 64%, p<0.0001). As expected, there was a higher prevalence of previous AMI or PCI, hypertension, or hyperlipidemia among AMI patients with the highest prevalence among NSTEMI patients (Table 1). Smoking was more prevalent in the myopericarditis group (66%) than in the NSTEMI (45%, p<0.001) or STEMI patients (39%, p<0.001). ST-segment elevation on the electrocardiogram was observed on presentation in 96% of the STEMI patients with the rest developing these changes during hospitalization. ST-elevation was apparent in only 31% of the myopericarditis patients (p<0.001). ST-segment depressions were evident in 32.4% of the NSTEMI patients but in only a few patients in the myopericarditis group (4.7%, p<0.001).

CRP sampled on presentation showed markedly higher values in the myopericarditis patients than among the NSTEMI and STEMI patients (Table 1, p<0.001). Troponin sampled on presentation showed similar values in STEMI and NSTEMI patients with slightly lower values in the myopericarditis group (p<0.05).

Fig 2 is a scatter plot of CRP vs. troponin values. Scrutiny of the localization of patient points on the CRP-troponin scatterplot showed a pattern of disparate distribution of points according to etiology. Those signifying myopericarditis patients were dispersed along the CRP-axis but usually at relatively lower troponin values. Points representing STEMI patients localize along and close to the troponin axis since these points were generally characterized by low CRP values and comparatively higher troponin values. Points representing NSTEMI

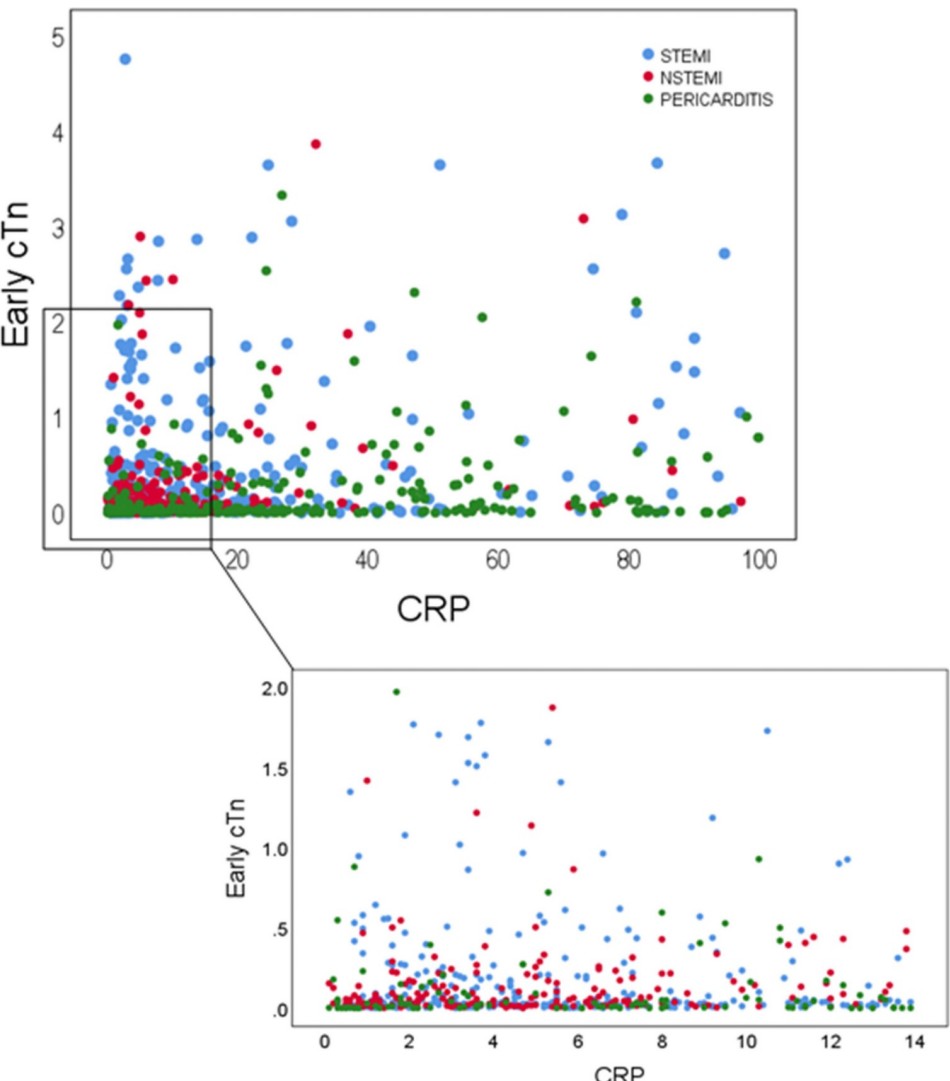

**Fig 2. Scatterplot of CRP vs. troponin on admission for study groups showing disparate point distribution.** Green circles signifying perimyocarditis patients are dispersed along the CRP axis at low troponin values. Points denoting STEMI patients localize typically along and closer to the troponin axis (blue circles) generally characterized by lower CRP values. Inset displays magnification of point scatter adjacent to origin. Red points representing NSTEMI patients are mostly characterized as expected by low CRP and troponin levels. cTn- cardiac troponin; CRP- C-reactive protein; STEMI- ST-elevation myocardial infarction; NSTEMI- Non STEMI.

patients were scattered in between, along the first half of the CRP axis at troponin values that were generally lower than troponin levels among STEMI patients.

Median CRP/troponin ratios sampled on presentation yielded significantly higher values in the myopericarditis patients than in both the STEMI and NSTEMI groups ($p<0.0001$, Table 1). There was a clear distinction of CRP/troponin ratios between study groups showing a divergence between the STEMI and the NSTEMI groups and the myopericarditis group (Fig 3, Table 1, $p<0.0001$).

The third and fourth quartiles of CRP/troponin ratio compared to the first quartile were statistically significantly associated with myopericarditis ($p = 0.005$, $p<0.0001$ respectively). The second to fourth age quartiles compared with the first quartile were negatively associated

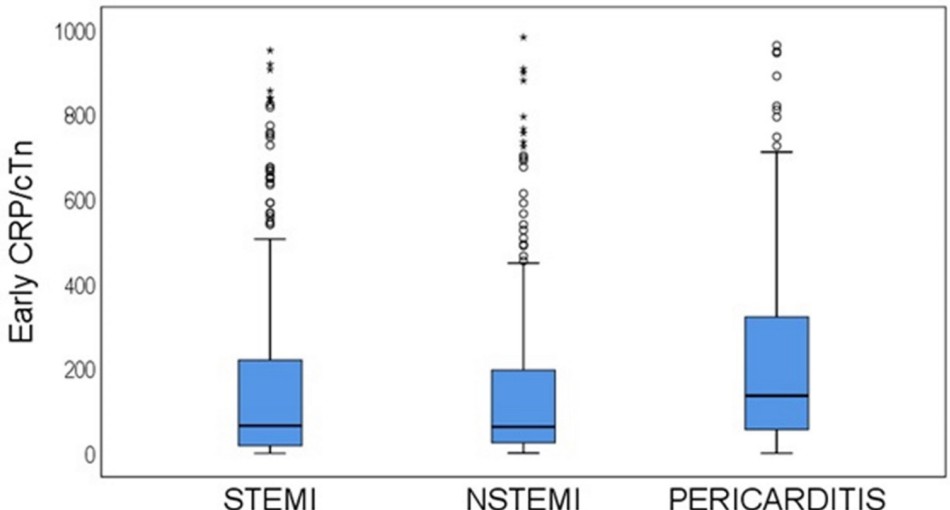

**Fig 3. Box plot of CRP/troponin ratio for the 3 study groups after exclusion of outliners beyond a limit of 750.**
There is a clear distinction between the perimyocarditis group and all patients (STEMI or NSTEMI). CTR- CRP/
troponin ratio; other abbreviations are the same as in Fig 2.

with myopericarditis (all p<0.0001). That is, the younger the patient, the more likely was a diagnosis of myopericarditis. There was no correlation between creatinine concentration and the diagnosis of myopericarditis (p = 0.52).

The ROC for the CRP/troponin ratio sampled on admission for diagnosis of myopericarditis against all AMI patients, i.e., both STEMI and NSTEMI patients, yielded an area under the curve (AUC) of 0.74 (CI: 0.7–0.77, p<0.0001, Fig 4A) with a Youden index of 501, and against STEMI patients yielded an AUC of 0.73 (CI: 0.69–0.76, p<0.0001) with a Youden index of 513 (Fig 4B). The ROC for the diagnosis of myopericarditis against NSTEMI patients yielded a slightly higher AUC (0.765, CI: 0.726–0.8, p<0.0001, Fig 4C). Fig 5 is a bar graph showing the prevalence of myopericarditis within deciles of the CRP/troponin ratio as percentage of total study population or as percentage of the patients within each CRP/troponin ratio decile. It

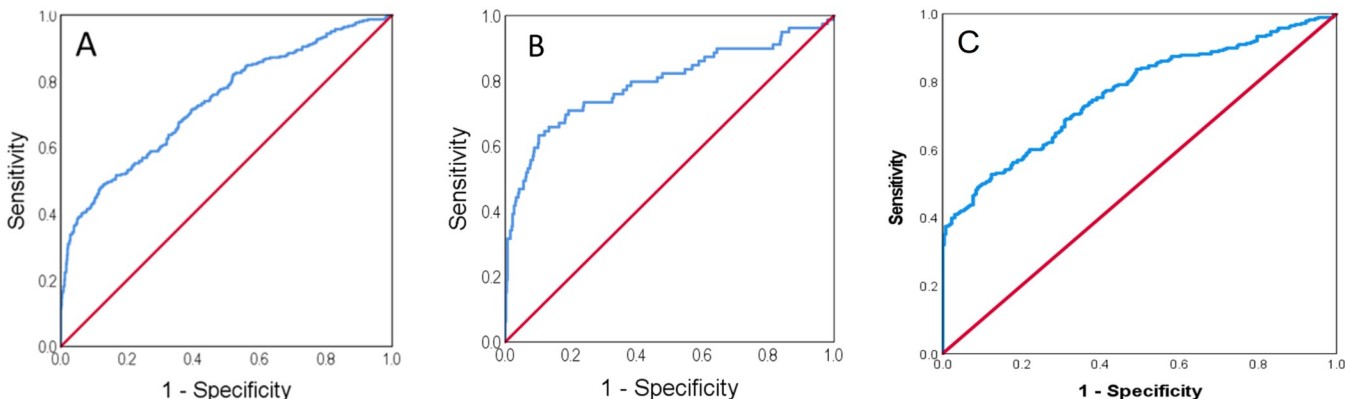

**Fig 4. The receiver-operating curve of the admission CRP/ troponin ratio for the diagnosis of perimyocarditis against STEMI and NSTEMI, or all acute myocardial infarction patients (A, AUC- 0.74, CI: 0.7–0.77, p<0.0001) or against STEMI patients only (B, AUC- 0.73, CI: 0.69–0.76, p<0.0001).**
Figure abbreviations are the same as in Fig 2.

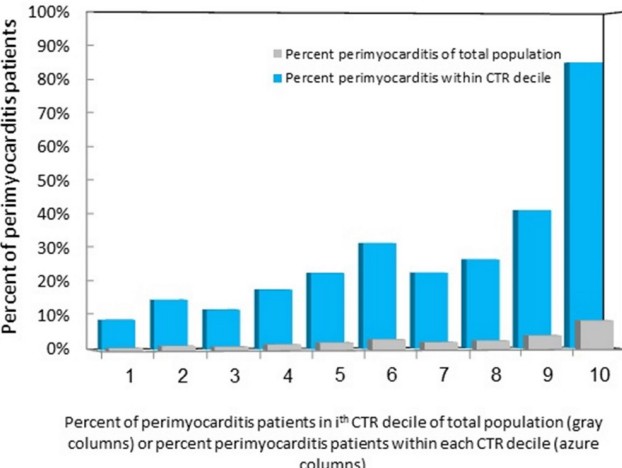

**Fig 5. Bar graph of the prevalence of the diagnosis of perimyocarditis within the CTR deciles as percentage of total study population (gray) or as percentage of patients within each CTR decile (azure).** It can be appreciated that as the CTR increases the percentage of perimyocarditis patients increases and so is its specificity. CTR- CRP/troponin ratio.

clearly shows that with increasing CRP/troponin ratio the percentage of myopericarditis patients rises and so is its specificity.

Calculation of the positive and negative likelihood ratios of the CRP/troponin ratio>500 to yield a true diagnosis of myopericarditis in the study population including all myocardial infarction patients yielded 3.8 and 0.5, respectively. The positive and negative likelihood ratios of the CRP/troponin ratio>500 to diagnose myopericarditis among all study patients yielded 3.4 and 0.58 values, respectively.

A similar calculation of positive and negative likelihood ratios at CRP/troponin = 500 for myopericarditis against NSTEMI only yielded 4.96 and 0.56, and against STEMI only yielded 3.4 and 0.58, respectively. By the same token, the positive and negative likelihoods for a CRP/troponin ratio>1000 to diagnose myopericarditis against AMI were 6.6 and 0.64, against STEMI were 5.1 and 0.65, and against NSTEMI 14.7 and 0.62, respectively.

## Discussion

The present study was designed to address the clinically relevant objective of assessing whether the CRP/troponin ratio obtained on patient admission to the hospital may aid in the differentiation between myopericarditis and STEMI (see central illustration). We reasoned that the CRP/troponin ratio, if found to be discriminative, could improve triage to the cardiac care unit or the cath lab for urgent PCI. In order to cover all options, we have analyzed the ability of CRP/troponin ratio on admission to differentiate myopericarditis patients from all AMI patients as well as from STEMI or NSTEMI patients only. We have shown in the present study, based on a registry of 1898 prospectively enlisted AMI and myopericarditis patients, that the use of admission CRP/troponin ratio serves as an aid in the triage of AMI patients by differentiating myopericarditis from AMI. The CRP/troponin ratio was characterized by a high negative predictive value along a large range of its value (Table 2). These results demonstrate a potential value for a high CRP/troponin ratio to support the diagnosis of myopericarditis in patients presenting with an acute cardiac syndrome when the clinical circumstances suggest low pre-test probability of obstructive coronary artery disease (Table 2).

**Table 2. The accuracy of CRP/troponin ratio levels measured early on presentation in STEMI with or without NSTEMI patients for distinguishing myopericarditis from myocardial infarction.**

|  | CRP/troponin ratio | Sensitivity | Specificity | PPV | NPV |
|---|---|---|---|---|---|
| CRP/Troponin in perimyocarditis vs. STEMI and NSTEMI' | **501**[*] | 50.0 | 86.9 | 60.0 | 81.5 |
|  | 30 | 88.9 | 30.9 | 33.6 | 87.6 |
|  | 60 | 84.7 | 44.9 | 37.7 | 88.2 |
|  | 100 | 75.3 | 56.0 | 40.3 | 85.2 |
|  | 150 | 69.1 | 63.7 | 42.8 | 84.0 |
|  | 300 | 56.3 | 76.6 | 48.6 | 81.7 |
|  | 1000 | 39.9 | 94.0 | 72.3 | 79.9 |
| CRP/Troponin in myopericarditis vs. STEMI | **513**[*] | 49.7 | 85.7 | 67.8 | 73.7 |
|  | 30 | 88.9 | 30.6 | 43.8 | 81.9 |
|  | 60 | 84.7 | 43.9 | 47.8 | 82.5 |
|  | 100 | 75.3 | 53.7 | 49.8 | 78.2 |
|  | 150 | 69.1 | 61.4 | 52.1 | 76.6 |
|  | 300 | 56.3 | 74.3 | 57.0 | 73.6 |
|  | 1000 | 39.9 | 92.2 | 75.7 | 71.6 |
| CRP/Troponin in myopericarditis vs. NSTEMI | **414**[*] | 52.8 | 87.6 | 82.6 | 62.4 |
|  | 30 | 88.9 | 34.1 | 59.1 | 71.7 |
|  | 60 | 84.7 | 46.9 | 64.0 | 73.3 |
|  | 100 | 75.3 | 30.1 | 67.8 | 68.6 |
|  | 150 | 69.1 | 67.8 | 70.6 | 66.3 |
|  | 300 | 56.3 | 81.0 | 76.8 | 62.4 |
|  | 1000 | 39.9 | 97.3 | 94.3 | 59.2 |

Asterisks denote Youden values.

The primary pathologic process in myopericarditis is inflammation associated with high CRP and usually relatively low-range, though higher than normal, troponin concentrations. In contrast, myocardial infarction is characterized by elevated troponin with a relatively lower increase in CRP levels. The combination of CRP and troponin has been sporadically evaluated jointly to evaluate the risk of cardiac disease and death in stable end-stage renal failure patients on dialysis [11] or to assess prognosis in acute coronary patients [12]. However, to the best of our knowledge, there is no study that evaluated the diagnostic value of the CRP/troponin ratio.

The scatter plot of CRP vs. troponin values shows a different localization of patients by diagnosis within the plot area (Fig 2). This 2-dimensional distribution underscores the capacity of the CRP/troponin ratio to aid in the discrimination between AMI and myopericarditis.

The calculation of the Youden J statistic provides an optimal value reconciling specificity and sensitivity. We, however, focused in the current analysis on specificity following the intention to reduce the rate of false positive events, namely, the number of patients admitted due to myocardial infarction but diagnosed and treated as myopericarditis. Consequently, a rather high CRP/troponin ratio, signifying high specificity, was selected as threshold for the diagnosis of myopericarditis in order to obtain a value beyond which the probability of active cardiac ischemia would be low. Hence, a high CRP/troponin value may assist in a tentative clinical diagnosis supporting a decision to avoid unnecessary diagnostic investigations or urgent procedures, frequently invasive in character. It is our opinion that implementation of the suggested CRP/troponin ratio criterion in the present population could probably have rendered most coronary angiograms performed in the myopericarditis patient subgroup included in the

present study unnecessary. A box plot of the CRP/troponin ratios of study groups shows a clear distinction between groups (Fig 3). The results suggest that a high CRP/troponin ratio, e.g., ratio>500 corresponds to specificity of >85% for myopericarditis (Table 2), and may preclude the need for coronary angiography. The ROC of the CRP/troponin ratio on presentation for diagnosis of myopericarditis against all AMI patients or against STEMI patients only yielded similar results (AUC = 0.74 or 0.73, p<0.0001, Fig 4A and 4B), but a slightly higher area against NSTEMI patients (AUC = 0.765, CI: 0.726–0.8, p<0.0001, Fig 4C), all demonstrating a good differentiating capability. The prevalence of myopericarditis diagnosis within deciles of the CRP/troponin ratio as percentage of the total study population or as patient percentage within each CRP/troponin ratio decile rises with increasing CRP/troponin ratio and so is its specificity (Fig 5).

It seems plausible that the CRP/troponin ratio may serve as an additional biomarker tool to use in clinical practice in order to evaluate the probability of myopericarditis. Though requiring further assessment, it may be that a CRP/troponin ratio>250 could support a probable diagnosis of myopericarditis, whereas a ratio>500 can substantiate this diagnosis as definite or highly probable (Table 2). High negative predictive values, even at lower CRP/troponin ratios, represent the other side of the coin, showing that though the vast majority of AMI patients present with low CRP/troponin ratios, there are some myopericarditis patients with lower ratios, simulating a myonecrosis process, which raises the need to exclude active myocardial ischemia in these patients. We have realized that most myopericarditis patients most resembled NSTEMI by clinical presentation and electrocardiographic pattern. Therefore, the most important aspect was to compare the diagnostic yield of the CRP/troponin ratio to distinguish between these entities. The ratio demonstrated a very high specificity and positive likelihood values for a ratio>500–1000, which are the important indicators toward the goal of minimizing the incidence of FP diagnosis.

## Limitations

Though possible bias in this retrospective registry-based study should be considered, the fact that consecutive patients were recorded prospectively virtually eliminates such an option. In the present registry maximal troponin values were reached in myocardial infarction patients either on the day of presentation or on the following day, whereas CRP in myopericarditis patients usually increased significantly on the second and third day of hospitalization (8). Unfortunately, these values are usually not available in many patients. If available, their ratio would have increased the disparity between the CRP/troponin ratio of study groups and its diagnostic accuracy. The design of the present study was based on the practiced routine in many hospitals, and represents a pragmatic protocol, which we have found useful for the clinical differentiation between AMI and myopericarditis.

In conclusion, we have shown in this study that the admission CRP/troponin ratio has the potential to assist in the discrimination between the phenotypically similar entities of AMI and myopericarditis.

## Supporting information

**S1 File.**
(DOCX)

## Author Contributions

**Conceptualization:** Simcha R. Meisel, Rami Abu Fanne, Ofer Kobo.

**Data curation:** Hamuda Nashed, Randa Natour, Naama Amsalem, Carmel Levin.

**Formal analysis:** Simcha R. Meisel, Rami Abu Fanne.

**Investigation:** Simcha R. Meisel, Majdi Saada, Avraham Shotan.

**Methodology:** Simcha R. Meisel, Majdi Saada.

**Resources:** Naama Amsalem.

**Supervision:** Simcha R. Meisel, Aaron Frimerman, Jameel Mohsen, Avraham Shotan, Ariel Roguin.

**Validation:** Simcha R. Meisel, Naama Amsalem, Jameel Mohsen, Avraham Shotan, Ariel Roguin, Michael Kleiner-Shochat.

**Visualization:** Michael Kleiner-Shochat.

**Writing – original draft:** Simcha R. Meisel.

**Writing – review & editing:** Rami Abu Fanne, Majdi Saada, Ofer Kobo, Aaron Frimerman, Yaniv Levi, Ariel Roguin.

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
