## [Decision Letter · Decision Letter 0]

8 Jan 2021

PONE-D-20-40200

Differentiation between Myopericarditis and Acute Myocardial Infarction on Presentation in the Emergency Department Using the Admission C-reactive Protein to Troponin Ratio

PLOS ONE

Dear Dr. Meisel,

Thank you for submitting your manuscript to PLOS ONE. After careful consideration, we feel that it has merit but does not fully meet PLOS ONE’s publication criteria as it currently stands. Therefore, we invite you to submit a revised version of the manuscript that addresses the points raised during the review process.

All issues raised by expert reviewers are required.

We look forward to receiving your revised manuscript.

Kind regards,

Vincenzo Lionetti, M.D., PhD

Academic Editor

PLOS ONE

Journal Requirements:

2. Please include in your Methods section (or in Supplementary Information files) the participating hospitals/institutions.

In the Methods section, please specify the source of the socio-demographic data that were used for this study.

Reviewers' comments:

Reviewer's Responses to Questions

**Comments to the Author**

1. Is the manuscript technically sound, and do the data support the conclusions?

Reviewer #1: Yes

Reviewer #2: Yes

2. Has the statistical analysis been performed appropriately and rigorously? 

Reviewer #1: Yes

Reviewer #2: Yes

3. Have the authors made all data underlying the findings in their manuscript fully available?

Reviewer #1: No

Reviewer #2: Yes

4. Is the manuscript presented in an intelligible fashion and written in standard English?

Reviewer #1: Yes

Reviewer #2: Yes

5. Review Comments to the Author

Reviewer #1: This is a single-center, retrospective study evaluating the C-reactive protein (CRP) to troponin ratio on presentation in 1898 patients admitted to the Emergency Department for chest pain. The Authors evaluated the performance of the CRP/troponin ratio alone to distinguish between myocardial infarction (MI; 81% of patients) and myopericarditis (19%). Median ratio values were much higher in patients with myopericarditis (p<0.001), with a fair performance at receiver operating characteristic analysis (area under the curve of 0.74), and a specificity >85% for a CRP/troponin ratio >500 (identified through the Youden method). The Authors conclude that the CRP/troponin ratio may allow to effectively discriminate between myopericarditis and MI. This conclusion is very reasonable, and may be seen as a way to reappraise the current diagnostic workup of chest pain, which includes the search for a dynamic troponin elevation and the assessment of inflammatory biomarkers, including CRP, as part of a broader evaluation. The identification of a rule-out cut-off for MI is particularly interesting in order to avoid further testing including urgent coronary angiography. In my opinion, this last point should be explored in a greater detail. Please find some other comments below.

The negative predictive value (NPV) and negative likelihood ratio (LR-) are important parameters for rule-out cut-off. Please provide their values for the 500 cut-off and check if you can find other cut-offs that maximize the NPV or with a LR- <0.1.

After identifying a rule-out cut-off in the whole population, please assess its performance across different age categories, given that patients with MI are usually older than those with myopericarditis (as confirmed in your cohort).

You should also assess the diagnostic performance of the troponin/CRP ratio to distinguish myopericarditis from NSTEMI alone.

Please evaluate also the diagnostic performance of your ratio across categories of renal function, which is known to influence troponin values.

Troponin values should be better expressed as ng/L. Furthermore, please specify throughout the paper that you are evaluating high-sensitivity troponin T.

To better quantify the added value of the troponin/CRP ratio, you may consider measuring the added value of the troponin/CRP ratio over troponin or CRP alone, or a combination of variables (for example, ischemic changes on ECG, history of coronary artery disease, typical chest pain, etc.) through discrimination analysis.

Please specify if the diagnosis of MI was based on European Society of Cardiology guidelines.

Reviewer #2: This is an interesting study aimed to assess the diagnostic role of CRP/troponin ratio to differentiate between myopericarditis and acute myocardial infarction. The Authors observed that high values of CRP/troponin ratio may support the diagnosis of myopericarditis in patients presenting with an acute cardiac syndrome. Data are convincing and correctly analyzed and discussed, however since the echocardiographic pattern of global hypokinesia is a strong predictor of myocarditis, it should be more useful to include in multivariate analysis also the presence of global hypokinesia. It is possible to hypotesize that CRP/Troponin ratio may be more useful to discriinate between ACS and myocarditis only in the subset of patients presenting with regional LV dysfunction and without pericarditis

6. PLOS authors have the option to publish the peer review history of their article (what does this mean?). If published, this will include your full peer review and any attached files.

Reviewer #1: **Yes: **Alberto Aimo

Reviewer #2: **Yes: **no

---

## [Author Response · Author response to Decision Letter 0]

18 Feb 2021

Reviewer #1: This is a single-center, retrospective study evaluating the C-reactive protein (CRP) to troponin ratio on presentation in 1898 patients admitted to the Emergency Department for chest pain. The Authors evaluated the performance of the CRP/troponin ratio alone to distinguish between myocardial infarction (MI; 81% of patients) and myopericarditis (19%). Median ratio values were much higher in patients with myopericarditis (p<0.001), with a fair performance at receiver operating characteristic analysis (area under the curve of 0.74), and a specificity >85% for a CRP/troponin ratio >500 (identified through the Youden method). The Authors conclude that the CRP/troponin ratio may allow effective discrimination between myopericarditis and MI. This conclusion is very reasonable, and may be seen as a way to reappraise the current diagnostic workup of chest pain, which includes the search for a dynamic troponin elevation and the assessment of inflammatory biomarkers, including CRP, as part of a broader evaluation. The identification of a rule-out cut-off for MI is particularly interesting in order to avoid further testing including urgent coronary angiography. In my opinion, this last point should be explored in a greater detail. Please find some other comments below.

The negative predictive value (NPV) and negative likelihood ratio (LR-) are important parameters for rule-out cut-off. Please provide their values for the 500 cut-off and check if you can find other cut-offs that maximize the NPV or with a LR- <0.1.

We have calculated the positive and negative likelihood ratios for the different subgroups as requested by the reviewer and have incorporated these calculations at the end of the results section. However, our opinion is that the indicators that should be optimized are the specificity, PPV, and possibly the positive LR. In other words, we should strive to minimize FP diagnosis of myopericarditis. This is because the consequence of a false-positive diagnosis is that an ischemic patient is considered to have myopericarditis, the result of which is inappropriate therapy (anti-inflammatory therapy rather than anti-aggregant and an invasive evaluation).

Calculation of the positive and negative likelihood ratios of the CRP/troponin ratio>500 to yield a true diagnosis of myopericarditis among all study patients yielded 3.8 and 0.5, respectively. A similar calculation of positive and negative likelihood ratios at CRP/troponin≥500 for myopericarditis against NSTEMI only yielded 4.96 and 0.56, and against STEMI only yielded 3.4 and 0.58, respectively. By the same token, the positive and negative likelihoods for a CRP/troponin ratio>1000 to diagnose myopericarditis against AMI were 6.6 and 0.64, against STEMI these were 5.1 and 0.65, and against NSTEMI- 14.7 and 0.62, respectively. Finally, we have realized that myopericarditis most resembled NSTEMI by clinical presentation and electrocardiographic pattern and, therefore as remarked by the reviewer, the most important aspect was to compare the diagnostic yield of the CRP/troponin ratio to distinguish between these entities. We have added all relevant statistical indicators (see additional values in Table 2) and found very high specificity and positive likelihood values for a ratio>500-1000, which are the important indicators as mentioned above (minimizing the incidence of FP). 

After identifying a rule-out cut-off in the whole population, please assess its performance across different age categories, given that patients with MI are usually older than those with myopericarditis (as confirmed in your cohort).

As a response to the comment by the reviewer a multivariate logistic regression model was performed in order to predict myopericarditis disease with adjustment to the independent parameters of age, CRP/troponin ratio, and creatinine. Patient age and CRP/troponin ratio were assessed in the model by their quartiles: <50, 50-59, 59-60, >69 years for age, and 29, 29-117, 117-455, >455 for the CRP/troponin ratio.

The second to the fourth age quartiles compared with the first quartile were negatively associated with myopericarditis (all p<0.0001). The younger the patient, the more likely was a diagnosis of myopericarditis. The third and fourth quartiles of CRP/troponin ratio compared with the first quartile were statistically significantly associated with myopericarditis (p=0.005, p<0.0001 respectively). There was no correlation between creatinine concentration and myopericarditis (p=0.52).

You should also assess the diagnostic performance of the troponin/CRP ratio to distinguish myopericarditis from NSTEMI alone.

We must admit that this requirement, as suggested by the reviewer, fits very well the character of the findings in this study, since the vast majority of myopericarditis patients presented with an ECG pattern and echocardiographic findings compatible with a NSTEMI and not as STEMI patients. We have, therefore, evaluated the accuracy of CRP/troponin (sensitivity, specificity, PPV, and NPV) to differentiate myopericarditis from NSTEMI (revised Table 2). We have also added to the revised paper the appropriate ROC figure not presented in the original version of the paper. In addition, we have evaluated the positive and negative likelihood ratios for CRP/troponin ratio=500 and 1000 to differentiate between myopericarditis and NSTEMI yielding the ratios of 4.96 and 0.56, respectively. These values are frequently encountered in these patients with myopericarditis (e.g. troponin- 0.04 ng/ml or [40 ng/liter] or 80 ng/liter and CRP- 40 mg/liter yielding a CRP/troponin of 1000 and 500, respectively). We have added to the conclusion section this consideration as well as the conversion factor for the sake of those centers who express troponin as ng/liter. At any rate, it is the novel concept that we describe and matters while every center can choose its ratio and to be applied in the clinical arena.

Please evaluate also the diagnostic performance of your ratio across categories of renal function, which is known to influence troponin values.

This is a good point and the comment makes sense, as the reviewer has probably assumed, that increased creatinine level representing lower renal function would lead to increased troponin level in the absence of ischemia. This could attenuate the accuracy of the CRP/troponin ratio to diagnose myopericarditis by spuriously decreasing its value. 

We thank the reviewer for this comment, since mildly elevated troponin levels in the presence of decreased renal function could certainly reduce the diagnostic accuracy of the CRP/troponin ratio even in the absence of ischemia. However, we assumed that such an effect would be minor since troponin usually demonstrates a 2-3-fold increase in patients with mild to moderate kidney failure whereas CRP increases 10 to 20 fold in myopericarditis. So, this effect was not expected to invalidate the diagnostic value of the CRP/troponin ratio. Anyway, as a response to the reviewer's comments, we have performed a logistic regression analysis to evaluate the relative effect of age, renal function and CRP/troponin ratio on the diagnosis of myopericarditis (Second comment, see above). The probability of pericarditis decreased by 10% for every 1-year increase in age (p<0.001), while creatinine level did not affect the accuracy of the CRP/ratio to diagnose myopericarditis. There was no correlation between creatinine concentration and myopericarditis (p=0.52). We incorporated this analysis in the revised manuscript.

Troponin values should be better expressed as ng/L. Furthermore, please specify throughout the paper that you are evaluating high-sensitivity troponin T.

We have, indeed, switched in our medical center to the ng/L dimensions for troponin values several years ago. However, much of the data was expressed as ng/ml when we have initially assessed the value of CRP/troponin in the early stages of data analysis. In addition, we felt that it would be conceptually better to express the combined biomarker CRP/troponin as an integer or >1 values and not as a fraction if we would have used ng/liter dimensions. For example, in a typical case of an AMI, a CRP value of 3 mg/liter with a troponin value of 100 ng/liter would yield a CRP/troponin value of 0.03, or 0.003 if measured troponin value is 1000 ng/liter. We felt that the CRP/troponin values of 30 and 3 obtained if ng/ml values are used, would be better mentally assimilated in routine practice. Following reviewer’s remark we added to paper this consideration that obtained CRP/troponin biomarker ratio should be divided by 1000 if used in the clinical field when troponin is measured in ng/liter. 

To better quantify the added value of the troponin/CRP ratio, you may consider measuring the added value of the troponin/CRP ratio over troponin or CRP alone, or a combination of variables (for example, ischemic changes on ECG, history of coronary artery disease, typical chest pain, etc.) through discrimination analysis.

In response to the recommendation of the reviewer, we have performed a similar analysis ROC analysis to assess the accuracy of CRP or troponin to differentiate between myopericarditis and AMI. Using Troponin yielded an AUC of 0.41 (CI: 0.363-0.454). This represents a low differentiating power, in some way, a chance finding. CRP findings were different. For the whole study population the obtained AUC was 0.747, almost identical to performance of the CRP/troponin ratio. This makes sense since a patient that presents with considerable CRP elevation (e.g., 40 or 60 mg/liter) and a slightly elevated troponin concentration probably is afflicted by myopericarditis, and not AMI that presents with a normal or minimally elevated CRP. However, at the lower CRP levels the CRP/troponin performs better to differentiate myopericarditis from myocardial infarction.

When evaluating the diagnostic accuracy of the CRP/troponin ratio to identify myopericarditis against acute myocardial infarction it performed similarly to the diagnostic yield of high-level CRP. However, at low but higher than normal concentrations of CRP, the ratio was diagnostic and CRP levels were not. Therefore, the CRP/troponin ratio should be used across all CRP levels as advocated in the paper. 

The diagnostic accuracy of the CRP/troponin ratio to diagnose myopericarditis against AMI yielded an AUC of 0.723 (CI: 0.6-0.844, p<0.001). The diagnostic accuracy of the CRP/troponin ratio to diagnose myopericarditis against NSTEMI yielded an AUC of 0.752 (CI: 0.62-0.884, p<0.001). Whereas, the diagnostic accuracy of the CRP to diagnose myopericarditis against NSTEMI yielded an AUC of 0. 56 (CI: 0.46-0.655, p=0.3) and was not statistically significant at all.

Please specify if the diagnosis of MI was based on European Society of Cardiology guidelines.

Yes, the diagnosis of MI was based on European Society of Cardiology guidelines. We assumed this fact as granted and, therefore, did not mention it explicitly. We thank the reviewer for his comment and added this premise to the manuscript. 

Reviewer #2: This is an interesting study aimed to assess the diagnostic role of CRP/troponin ratio to differentiate between myopericarditis and acute myocardial infarction. The Authors observed that high values of CRP/troponin ratio may support the diagnosis of myopericarditis in patients presenting with an acute cardiac syndrome. 

Data are convincing and correctly analyzed and discussed, however since the echocardiographic pattern of global hypokinesia is a strong predictor of myopericarditis, it should be more useful to include in multivariate analysis also the presence of global hypokinesia. It is possible to hypothesize that CRP/Troponin ratio may be more useful to discriminate between ACS and myocarditis only in the subset of patients presenting with regional LV dysfunction and without pericarditis

Following analysis of the data, or the echocardiographic data in particular, we have observed that the classic pattern of diffuse hypokinesis is rare, especially if myopericarditis (and not myocarditis) patients are evaluated. In agreement with the reviewer we must admit that most myopericarditis patients present, with regard to their ECG pattern and echocardiographic findings and frequently also clinically, as NSTEMI and not as STEMI patients. Hence, as the reviewer stated, the diagnostic challenge is to differentiate myopericarditis patients with electrocardiographic STT changes and regional wall motion abnormalities resembling acute myocardial infarction from NSTEMI patients. Hence, we have reanalyzed the data after excluding the patients with no wall motion abnormalities or those with uniform wall motion abnormality. Seventy-seven patients (21.7%) of the myopericarditis group compared to 18% among the NSTEMI group and 8% among the STEMI group did not manifest regional wall motion abnormalities on echocardiography. Re-calculation of ROC after excluding this population yielded an AUC of 0.745, very similar to the result obtained for the entire study population regardless of their echocardiographic pattern. In other words, including the only patients, that resemble AMI patients, with regional motion abnormalities, did not significantly alter the accuracy of the CRP/troponin ratio for differentiating myopericarditis from AMI.

---

## [Decision Letter · Decision Letter 1]

25 Feb 2021

Differentiation between Myopericarditis and Acute Myocardial Infarction on Presentation in the Emergency Department Using the Admission C-reactive Protein to Troponin Ratio

PONE-D-20-40200R1

Dear Dr. Meisel,

We’re pleased to inform you that your manuscript has been judged scientifically suitable for publication and will be formally accepted for publication once it meets all outstanding technical requirements.

Kind regards,

Vincenzo Lionetti, M.D., PhD

Academic Editor

PLOS ONE

Additional Editor Comments (optional):

Reviewers' comments:

Reviewer's Responses to Questions

**Comments to the Author**

1. If the authors have adequately addressed your comments raised in a previous round of review and you feel that this manuscript is now acceptable for publication, you may indicate that here to bypass the “Comments to the Author” section, enter your conflict of interest statement in the “Confidential to Editor” section, and submit your "Accept" recommendation.

Reviewer #1: All comments have been addressed

Reviewer #2: All comments have been addressed

2. Is the manuscript technically sound, and do the data support the conclusions?

Reviewer #1: Yes

Reviewer #2: Yes

3. Has the statistical analysis been performed appropriately and rigorously? 

Reviewer #1: Yes

Reviewer #2: Yes

4. Have the authors made all data underlying the findings in their manuscript fully available?

Reviewer #1: Yes

Reviewer #2: Yes

5. Is the manuscript presented in an intelligible fashion and written in standard English?

Reviewer #1: Yes

Reviewer #2: Yes

6. Review Comments to the Author

Reviewer #1: The Authors have modified their manuscript according to my suggestions. I have no further comments.

Reviewer #2: I do not any further comments, the quality of paper is improved after revision. Major issues are solved

7. PLOS authors have the option to publish the peer review history of their article (what does this mean?). If published, this will include your full peer review and any attached files.

Reviewer #1: No

Reviewer #2: **Yes: **Luciano Agati

---

## [Editor Report · Acceptance letter]

6 Apr 2021

PONE-D-20-40200R1 

Differentiation between Myopericarditis and Acute Myocardial Infarction on Presentation in the Emergency Department Using the Admission C-reactive Protein to Troponin Ratio 

Dear Dr. Meisel:

I'm pleased to inform you that your manuscript has been deemed suitable for publication in PLOS ONE. Congratulations! Your manuscript is now with our production department. 

Kind regards, 

on behalf of

Prof. Vincenzo Lionetti 

Academic Editor

PLOS ONE